# Dengue vascular leakage is augmented by mast cell degranulation mediated by immunoglobulin Fcγ receptors

**Ayesa Syenina[1†], Cyril J Jagaraj[1†], Siti AB Aman[1], Aishwarya Sridharan[1], Ashley L St John[1,2]\***

[1]Program in Emerging Infectious Diseases, Duke-National University of Singapore, Singapore, Singapore; [2]Department of Pathology, Duke University Medical Center, Durham, United States

**Abstract** Dengue virus (DENV) is the most significant human arboviral pathogen and causes ~400 million infections in humans each year. In previous work, we observed that mast cells (MC) mediate vascular leakage during DENV infection in mice and that levels of MC activation are correlated with disease severity in human DENV patients (*St John et al., 2013b*). A major risk factor for developing severe dengue is secondary infection with a heterologous serotype. The dominant theory explaining increased severity during secondary DENV infection is that cross-reactive but non-neutralizing antibodies promote uptake of virus and allow enhanced replication. Here, we define another mechanism, dependent on FcγR-mediated enhanced degranulation responses by MCs. Antibody-dependent mast cell activation constitutes a novel mechanism to explain enhanced vascular leakage during secondary DENV infection.

**\*For correspondence:** ashley.st.john@duke-nus.edu.sg

[†]These authors contributed equally to this work

**Competing interests:** The authors declare that no competing interests exist.

## Introduction

Infection with dengue virus (DENV) can result in a wide range of clinical manifestations, from asymptomatic, to mild and self-limiting, to severe and occasionally fatal. Febrile disease that is self-resolving is diagnosed as dengue fever (DF). More severe forms of the disease include dengue hemorrhagic fever (DHF) and dengue shock syndrome (DSS), which are characterized by increased vascular permeability and plasma leakage into tissues (*Halstead, 2007*; *St John et al., 2013a*). There are four major serotypes of DENV that infect humans, and a primary DENV infection with any of the four serotypes elicits long-lived antibodies against the primary infection serotype (*St John et al., 2013a*). Importantly, this DENV-specific humoral response is capable of preventing infection by neutralizing virus of the infecting serotype only. Although the antibody response generated during a primary infection frequently shows some cross-reactivity to other DENV serotypes, these responses are largely non-neutralizing towards viruses of other DENV serotypes (*Halstead, 2007*; *Rothman, 2011*). Symptomatic re-infection with the same serotype that had been experienced during primary infection has never been reported; therefore, immunological memory is understood to be highly effective against a secondary infection with a homologous serotype (*Sabin, 1952*). In contrast, symptomatic secondary infection with a heterologous serotype of DENV is relatively common in DENV-endemic regions (*Low et al., 2006*; *Guilarde et al., 2008*; *Anderson et al., 2014*). Some studies have shown that there is an increased risk of developing DHF due to a secondary infection with a heterologous DENV serotype, or as a result of the presence of maternally transferred heterologous antibodies (*Halstead et al., 1970*; *Halstead, 1988*; *Kliks et al., 1988*).

One possible explanation for the increased risk of developing severe dengue during secondary heterologous infection is the process known as antibody-dependent enhancement of infection (ADE)

(*Halstead et al., 1973*; *Halstead and O'Rourke, 1977a*, *1977b*). It is thought that more severe infections are associated with higher levels of virus replication, which could be attributed to cellular uptake of immune complexes that are internalized by immunoglobulin Fc-receptor (FcR) bearing cells, such as dendritic cells and macrophages (*Kliks et al., 1988*). ADE-promoting immune complexes are generated when pre-existing, non-neutralizing antibodies cross-react with an antigenically distinct heterologous DENV serotype. These weak interactions allow virus binding to antibodies without killing the virus, followed by attachment of immune complexes to cells, which promotes uptake of virus. It is thought that increased viral infection as a result of ADE leads to increased secretion of vasoactive immune products by infected cells, such as TNF-α, which may promote vascular leakage during DHF (*Kliks et al., 1988*). Conflicting information regarding the association of TNF-α, specifically, with severe dengue has been reported (*Hober et al., 1993*; *Chakravarti and Kumaria, 2006*) but cytokines produced by DENV-infected cells remain the leading suspects for inducing severe dengue (*St John et al., 2013a*). This view of ADE emphasizes the role that viral replication plays in generating pro-inflammatory and vasoactive factors. Although ADE, alone, is not sufficient to explain the vascular pathology associated with DHF since many individuals also experience DHF during primary infection, it can explain antibody-dependent increases in viral uptake and infection burden that could promote downstream pathogenesis (*St John et al., 2013a*). Weakly neutralizing antibodies against the DENV structural proteins, including envelope (E) protein domain III and precursor membrane protein, have been observed in human DENV patients (*Dejnirattisai et al., 2010*; *Wahala et al., 2012*) and the phenomenon of ADE has been shown in vitro and in animal models (*Halstead et al., 1973*; *Shresta et al., 2006*); however, ADE has not yet been observed in human DENV patients (*St John et al., 2013a*). As a result, while many lines of evidence support the theory of ADE as a contributing factor to DENV severity, this does not exclude the possibility of additional mechanisms for antibodies to promote vascular pathology during DENV infection.

In addition to the standard pathway of uptake of antigen–antibody complexes and processing in cytosolic compartments, there are other unique pathways that the host uses to generate responses to antibody-bound antigens. Mast cells (MCs), for example, are granulated cells that express a wide range of Fc receptors (*Abraham and St John, 2010*). This allows them to bind multiple classes of antibodies, including IgE (through FcεRI) and IgG (through FcγRs) (*Sylvestre and Ravetch, 1996*). Human MCs have been shown to express the inhibitory IgG receptor, FcγRIIb, as well as the activating receptors, FcγRI and FcγRIIa (which are functionally analogous to FcγRIII on mice), albeit with some variations based on tissue and cell activation state (*Sylvestre and Ravetch, 1996*; *Okayama et al., 2000*; *Malbec and Daëron, 2007*). MCs are important for pathogen immunosurveillance and pre-store many vasoactive mediators within their granules (*Abraham and St John, 2010*). Recent reports showed that MCs, which are localized in tissues near the vasculature (*Kunder et al., 2011*), are directly activated by DENV, resulting in the release of their pre-formed mediators through the process of degranulation (*St John et al., 2011*, *2013b*). In our previous study, we showed that DENV-elicited MC vasoactive products enhance vascular leakage in animal models (*St John et al., 2013b*). We also reported that DHF patients with secondary infection, in contrast to DHF patients with primary infection, displayed higher serum levels of the MC-derived protein chymase, a biomarker that is specifically produced by MCs and associated with MC activation (*St John et al., 2013b*). This suggests that MC degranulation levels are higher in patients with secondary infection, raising the question of how MC responses could be enhanced, mechanistically, during secondary infection (*St John, 2013*).

IgG is efficiently bound by Fc-receptor bearing cells when antibody and antigen aggregate in immune complexes. Similarly to other Fc-receptor bearing cells, MCs also can be infected through the process of ADE when immune complexes containing non-neutralized DENV are taken up via Fc receptor-mediated endocytosis (*King et al., 2000*; *Goncalvez et al., 2007*). However, in addition to endocytosis of immune complexes, MCs act as effectors of the adaptive immune system due to their ability to become sensitized by pre-existing circulating antibodies through binding to the Fc receptors (*Abraham and St John, 2010*). When antibody is bound in immune complexes, affinity for the IgG receptors increases substantially (*Takizawa et al., 1992*). Antibody-sensitized MCs can become activated during a secondary challenge when the sensitizing-antigen binds to Ig and cross-links the FcRs or can be activated through binding of immune complexes that have already been formed (*Oshiba et al., 1996*; *Malbec and Daëron, 2007*). While the downstream effects of antibody-sensitization of MCs have been well-characterized in allergy and hypersensitivity models, there is still a lack of understanding of how this process could be relevant in the context of viral infection. In vitro,

it has been shown that MCs sensitized with post-immune serum containing IgE antibodies have increased degranulation in response to DENV (*Sanchez et al., 1986*). Interestingly, increased DENV-specific IgE antibody levels have been associated with DHF (*Koraka et al., 2003*). With regards to IgG-mediated MC activation, studies have also shown that IgG immune complexes can promote MC degranulation in mice (*Vaz and Ovary, 1968*; *Vaz and Prouvost-Danon, 1969*; *Daëron et al., 1992*). Moreover, mice deficient in FcγRIII lack IgG-mediated MC degranulation, have an impaired Arthus reaction, and are resistant to IgG-dependent passive anaphylaxis (*Hazenbos et al., 1996*). Based on our previous study that demonstrated that MCs are essential for maximal vascular leakage during DENV infection (*St John et al., 2013b*), we hypothesized that antibody-mediated enhancement of MC activation and degranulation could potentially increase vascular permeability in the associated blood vessels during secondary DENV infection. In this study, we sought to investigate the ability of DENV-specific IgG antibodies to mediate enhanced vascular leakage through MCs and determine the mechanism of this interaction.

## Results

### DENV-specific IgG enhances MC-degranulation and vascular leakage

To determine if MC degranulation is affected by antibody-mediated sensitization in vitro, we exposed MCs to a clinical isolate of DENV2, Eden2, after pre-treatment using increasing concentrations of the 4G2 antibody, an IgG2a monoclonal antibody that is cross-reactive for the envelope (E) protein of DENV serotypes 1–4 (*Sithiprasasna et al., 1994*). Degranulation was measured using a standard β-hexosaminidase assay. Consistent with our prior study (*St John et al., 2011*), DENV-induced antibody-independent MC degranulation in vitro (*Figure 1A*). However, we found that sensitization of MCs with 4G2 significantly increases the magnitude of MC degranulation in response to DENV2, compared to virus exposure alone (*Figure 1A*). Furthermore, increasing concentrations of antibody resulted in incrementally increased MC degranulation in a dose-dependent fashion (*Figure 1A*).

To assess if the enhancing effect of antibodies on DENV-induced MC degranulation could be observed in vivo, we performed a similar experiment in mice. Mice were pre-treated with antibody 4G2 by intra-peritoneal (i.p.) injection, 24 hr prior to infection with $1 \times 10^6$ pfu of DENV2 (also by i.p. injection). The DENV2 strain that we used for in vivo infection, Eden2, is a clinical isolate that sustains transient replicating infection in WT mice for approximately 5 days (*St John et al., 2013b*). The peritoneal cells, including peritoneal MCs, were isolated by at lavage 24 hr post-infection and degranulation was assessed by both flow cytometry (*Figure 1B*) and immunostaining (*Figure 1C*). Both techniques revealed that MC degranulation occurs in response to DENV alone (as previously reported [*St John et al., 2011*]) but that pre-treatment with the anti-E antibody 4G2 significantly enhances MC-degranulation in vivo over the levels induced by DENV exposure alone (*Figure 1B–C*). These data show that DENV-specific IgG antibodies have the potential to enhance MC responses in vivo.

We have previously reported that DENV-induced MC degranulation increases endothelial permeability in vitro, and vascular leakage in vivo during primary infection of naïve animals (*St John et al., 2013b*). Therefore, we hypothesized that the increased degranulation of MCs in the presence of pre-formed antibodies would also enhance the extent of vascular leakage during DENV infection. To determine if DENV-specific antibodies could enhance vascular leakage during DENV infection, we pretreated one group with the antibody 4G2 by i.p. injection, prior to infection with DENV, as described above. Vascular leakage was assessed 24 hr and 48 hr post-infection. To measure vascular leakage, we obtained blood from mice and acquired hematocrit values. Hematocrit is a measure of the percentage of red blood cells in the total blood volume, which is used as an indicator of vascular permeability (as a result of plasma loss from the circulation) in human DENV patients and in DENV mouse models (*World Health Organization, 2009*; *St John et al., 2013b*). Measurement of hematocrit levels at 24 hr and 48 hr post-DENV2 infection showed that mice that had been pre-sensitized to DENV by injection with 4G2 antibody prior to DENV2 infection had significantly increased hematocrit values compared to those that were infected with DENV in the absence of the antibody (*Figure 1D*). Interestingly, although the hematocrit values were increased in antibody pre-treated mice, the levels of virus were not significantly increased in the spleen (*St John et al., 2013b*), a DENV-target organ, (*Figure 1E*) at the same time point.

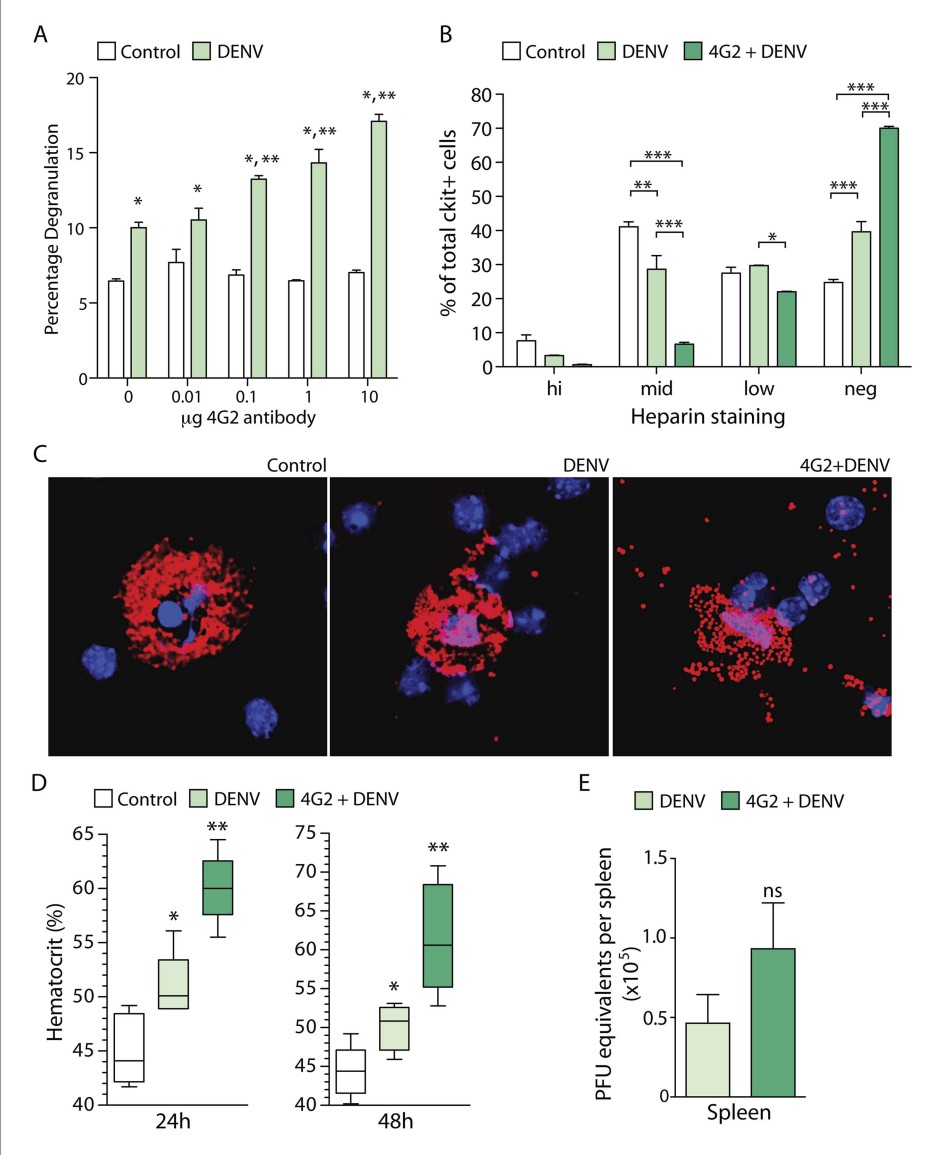

**Figure 1**. DENV-specific IgG enhances MC degranulation and vascular leakage. (**A**) MC-degranulation (cell line RBL-2H3) in response to DENV2 is enhanced by dose-dependent increases in the DENV E-specific antibody 4G2. Significance was determined by 2-way ANOVA, where * indicates a significant increase over control, while ** indicates a significant increase over DENV alone without 4G2 antibody. (**B**) In vivo degranulation of peritoneal MCs was enhanced by DENV compared to isotype controls and 4G2+DENV compared to both controls and DENV infection alone. MC degranulation was determined by flow cytometry to quantitate the heparin-containing granules as an assessment of granularity. Heparin staining of MCs (ckit+) was partitioned into high, mid, low, and negative populations (*St John et al., 2011*). DENV infection resulted in a significant loss of granulated MCs in the mid partition, and a significant increase in the percentage of cells in the heparin-negative partition (presumably degranulated). This was further accentuated by 4G2 treatment so that approximately 70% of ckit+ cells lacked heparin-staining granules in that group, while both mid and low heparin-staining groups were significantly reduced compared to DENV infection alone. Data were analyzed by 2-way ANOVA with Bonferroni's multiple comparison test to obtain p-values for between group comparisons: *$p < 0.05$, **$p < 0.001$, ***$p < 0.0001$. For each group, $n = 3$ animals and consistent data were obtained in a second independent trial. (**C**) Images of MCs were taken by confocal microscopy after cytospinning peritoneal lavage. Slides were stained for MC-heparin to reveal granules and DAPI for nuclei. Control MCs were densely granulated, but degranulation could be observed in samples from DENV-infected animals with and without 4G2 pre-sensitization. More granules appeared extracellular on slides in the 4G2+DENV group. (**D**) Vascular leakage was measured by obtaining hematocrit values 24 hr and 48 hr after infection with DENV, with and without pre-treatment with 4G2. Significance was determined by 1-way ANOVA with Bonferroni's post-test,

*Figure 1. continued on next page*

*Figure 1. Continued*

where * indicates a significant increase over control, while ** indicates a significant increase over both control and DENV alone. (**E**) PFU equivalents were measured in mice by real time RT-PCR using RNA isolated from spleens, 24 hr after infection with DENV. Means do not differ significantly by Student's un-paired *t*-test (p = 0.2073, n = 5).

## Enhanced degranulation and vascular leakage by serotype-specific and cross-reactive IgG

We next aimed to examine the contribution of antibody specificity against the DENV E protein to MC degranulation. For this, we analyzed the degranulation of DENV-exposed MCs after pre-treatment with either the serotype-cross-reactive antibody 4G2 or the DENV2-specific 3H5 antibody, an IgG1 monoclonal antibody specific for the E protein of DENV2 that does not cross-react with E from other DENV serotypes (*Henchal et al., 1982*). By β-hexosaminidase assay, we observed increased degranulation in the presence of the 4G2 antibody for all four serotypes of DENV, compared to DENV treatment alone (*Figure 2A*). In contrast, the DENV2-specific 3H5 antibody only enhanced DENV2-induced degranulation over DENV2 alone treatment, but did not significantly change the levels of degranulation induced by MC exposure to DENV1, 3, or 4 (*Figure 2B*). Therefore, specificity or cross-reactivity of antibody for the infecting DENV serotype is required for antibody-mediated enhanced MC degranulation in vitro.

Subsequently, we assessed the role of antibody specificity in vivo to MC responses to DENV through an experiment where mice were pre-sensitized (as described above) with either antibody 4G2 or 3H5, followed by a DENV1 challenge. 4G2 represents a heterologous antibody against DENV1 since it cross-reacts with DENV1, while 3H5 represents a non-specific antibody for DENV1, allowing us to assess the influence of IgG cross-reactivity to vascular leakage in vivo. Vascular permeability was measured using hematocrit values obtained from blood at the final endpoint (*Figure 2C*). We found that mice pre-sensitized with 4G2 prior to DENV1 infection had higher hematocrit values compared to those sensitized with control antibody, while 3H5 produced no enhancing effect during DENV1 infection (*Figure 2C*). We previously optimized a technique for measuring vascular leakage due to DENV in the WT mouse model, involving injection of Evan's blue dye (EBD) 30 min prior to euthanasia, followed by perfusion of the mouse vasculature with saline before tissue observation and harvest (*St John et al., 2013b*). This allowed the measurement of the EBD leakage into the liver by determining the OD-600 from the supernatants of homogenized liver tissue. Using this technique as a secondary method to assess vascular leakage quantitatively, we observed that while DENV1 alone increased vascular leakage over control values, the leakage was significantly enhanced in the presence of antibody 4G2 (*Figure 2D*). Again, in contrast to the DENV1–4 cross-reactive antibody 4G2, DENV2-specific 3H5 had no effect on vascular leakage when administered prior to a DENV1 challenge (*Figure 2D*). These quantitative results were also visually supported, as shown in *Figure 2E*, when mouse livers were imaged after i.v. EBD injection and perfusion of the circulatory system with saline. DENV1 infection alone (without antibody pre-treatment) appears to increase vascular leakage in the liver tissue over control, so that bruising remains on the liver even after the blood has been eliminated from the vasculature (*Figure 2E*). This vascular leakage is not apparent on the livers of uninfected control animals (*Figure 2E*). In contrast, the most visually striking vascular leakage occurred in animals pre-treated with 4G2 prior to DENV1 infection (*Figure 2E*). These findings support that the enhanced vascular leakage DENV induces in the presence of antibodies is dependent on antibody specificity to the infecting DENV-serotype.

## The role of MCs in IgG-enhanced vascular leakage

Having observed that the DENV2-specific antibodies promote increased MC degranulation and vascular leakage in infected WT mice, we wanted to investigate the contribution of MCs to the increased vascular pathology in the presence of a DENV-specific antibody. To identify the role of MCs, we compared vascular leakage between DENV-infected WT mice and MC-deficient mice (Sash) in the presence of 3H5 antibody. As before, mice were injected with 3H5 antibody 24 hr prior to infection with DENV2, and hematocrit levels were measured at 24 hr post-infection. Hematocrit analysis

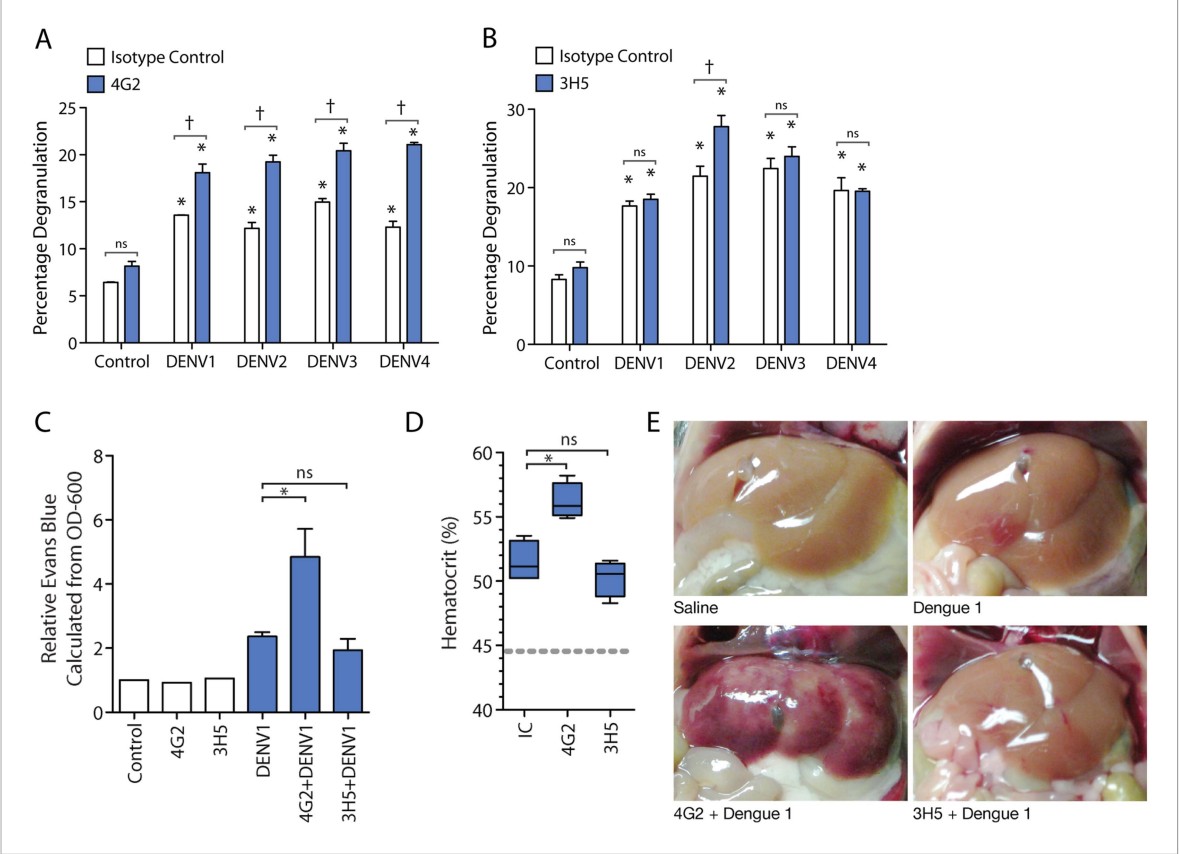

**Figure 2**. Antibody specificity governs IgG-enhanced MC degranulation and vascular leakage. (**A**) MC degranulation in response to DENV1–4 is enhanced by cross-reactive antibody 4G2. (**B**) DENV2-specific antibody 3H5 enhances MC degranulation in response to DENV2 but not DENV1, 3 or 4. For **A** and **B**, data were analyzed by 2-way ANOVA with Bonferroni's post-test to determine p-values for group comparisons; * indicates a significant enhancement over control (p < 0.05), † indicates a significant increase over DENV treatment with isotype control (IC) for a given serotype (p < 0.05), and ns designates 'not significant'. (**C**) Hematocrit values and (**D**) Evan's blue dye leakage into tissues were measured 24 hr after DENV1 infection of mice that had been pretreated with IC, 4G2, or 3H5. Hematocrit values and EBD detection values were elevated over baseline for all DENV1-infected mice; however, only the cross-reactive antibody 4G2 increased hematocrit values and Evans blue dye leakage over DENV1 infection with IC antibody. For **C** and **D**, significance was determined by 1-way ANOVA with Bonferroni's post-test; p < 0.05. (**E**) Images show the appearance of mouse livers after saline perfusion. For mice given control injections of saline, the perfusion results in complete flushing of the blood and EBD from the liver. In contrast, bruising consistent with vascular leakage could be observed with DENV1 alone infection, which appears enhanced in mice pre-treated with DENV1 cross-reactive antibody 4G2, but similar in animals pre-treated with non-binding antibody 3H5.

supported MC-dependent antibody-enhanced vascular leakage since WT mice had substantially higher hematocrit values during DENV2 infection in the presence of DENV2-specific antibodies, while MC-deficient Sash mice showed no changes in hematocrit over baseline controls for either DENV2 treatment alone or treatment with 3H5 and DENV2 (*Figure 3A*). These results were also supported using the secondary method of quantitating vascular leakage into tissues by measuring EBD leakage into tissues (*Figure 3A*). While DENV2 induced significantly increased vascular leakage, the DENV2-specific antibody 3H5 further increased vascular leakage compared to both baseline and DENV2 infection alone (*Figure 3A*). We have reported previously that during DENV infection in wild type, immunocompetent mice, vascular perfusion with saline was required to visualize EBD and plasma leakage into highly vascularized tissues such as the liver and kidney (*St John et al., 2013b*). Surprisingly, when enhancing antibodies were administered 24 hr prior to infection with DENV, the resulting experimental outcome was strong enough that we were able to observe overt leakage of EBD on the gut post-infection during necropsy. In contrast, this overt increase in vascular leakage was not apparent in Sash mice that had similarly been pretreated with 3H5 followed by DENV2 infection (*Figure 3B*). This observation that MCs are required for increased DENV-induced, antibody-enhanced

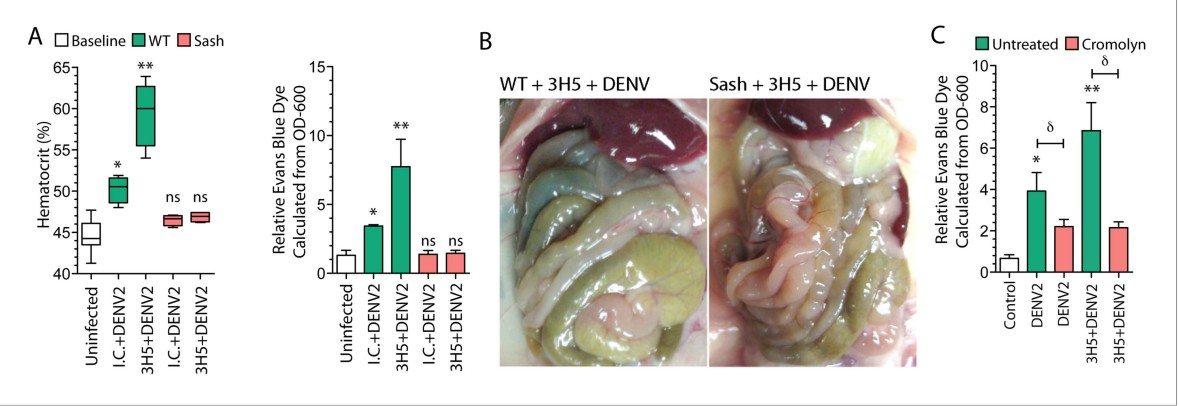

**Figure 3**. Antibody-enhanced vascular leakage is MC dependent. (**A**) WT and Sash mice were given IC antibody or 3H5 injections 24 hr prior to infection with DENV2. At 24 hr post-infection, blood was obtained for hematocrit analysis. WT mice given DENV2 after either IC or 3H5 treatment had significantly elevated hematocrit values compared to uninfected controls while hematocrit values of Sash mice were not significantly (ns) elevated over controls. EBD in the liver tissue was measured at 24 hr after infection and 30 min after injection with EBD. (**B**) Images showing the organs in the peritoneal cavity, taken prior to saline perfusion, from necropsy of representative WT and Sash mice with 3H5-antibody-enhanced DENV2 infection showing the organs in the peritoneal cavity. EBD can be visually observed in the gut of WT mice but not in Sash mice. (**C**) WT mice with and without 3H5 pretreatment before DENV2 infection were treated with the MC stabilizer, Cromolyn. Vascular leakage was measured by EBD perfusion 24 hr post-infection, followed by quantitation in liver tissue. For panels (**A–C**), significance was determined by 1-way ANOVA with Bonferroni's post-test; * indicates a significant increase over control and ** indicates a significant increase over control as well as the DENV2-infected group that was not pre-treated with 3H5. For (**C**), δ indicates a significant reduction in vascular leakage after Cromolyn treatment, compared to the untreated group.

vascular permeability, although qualitative in *Figure 3B*, was consistent with the significantly increased vascular leakage in the presence of antibody that was quantitated in *Figure 3A*.

Since antibody-enhanced MC degranulation to DENV occurs in vivo (*Figure 1*) and the resulting increases in vascular leakage in our model appear to be MC dependent (*Figure 3A–B*), we sought to determine if pharmacological suppression of the responses of MCs could limit antibody-enhanced vascular leakage during DENV infection. For this, we treated the mice that had been infected with DENV using the MC stabilizing drug, cromolyn, which blocks MC degranulation. Cromolyn treatment successfully reduced both DENV-induced vascular leakage, alone, as well as antibody-enhanced DENV vascular leakage (*Figure 3C*).

## FcγRIII receptors mediate IgG-enhanced MC responses

Since we found that the IgG monoclonal antibodies 4G2 and 3H5 promote MC dependent increased vascular leakage in a DENV serotype-specific manner, we next investigated whether Fc receptors on MCs are involved in IgG-enhanced MC degranulation and increased vascular permeability. There are two activating Fcγ receptors, FcγRI and FcγRIIa, which are expressed on human MCs that bind the Fc regions of IgG antibodies. Although FcγRI could theoretically contribute to MC-degranulation in humans (*Malbec and Daëron, 2007*), it has not been identified on mouse MCs, and thus, is not expected to be involved in the mechanism we observed. Therefore, we hypothesized that FcγRIII, which serves a similar activating function on mouse MCs, could be primarily involved in DENV-specific IgG-enhanced MC degranulation. To test this hypothesis, bone marrow-derived MCs (BMMCs) were generated from either WT mice or mice deficient in FcγRIII (FcγRIII-KO). Prior to performing a standard β-hexosaminidase assay, cells were incubated with the antibodies 4G2 or 3H5, followed by stimulation with DENV2. BMMC degranulation responses to DENV were significantly enhanced in WT BMMCs pre-treated with either antibody, while the antibody-enhanced response was abrogated by FcγRIII deficiency (*Figure 4A*). No difference was observed for direct degranulation responses to DENV in FcγRIII-KO BMMCs compared to WT BMMCs (*Figure 4A*). Similarly, degranulation of WT and FcγRIII-KO BMMCs was assessed in response to DENV1, after exposure of MCs to either DENV1-non-specific antibody, 3H5, or heterologous antibody, 4G2 (*Figure 4B*). As expected, only 4G2, and not 3H5, enhanced WT BMMC degranulation responses to DENV1, but this 4G2-dependent enhancement was not observed in FcγRIII-KO BMMCs (*Figure 4B*).

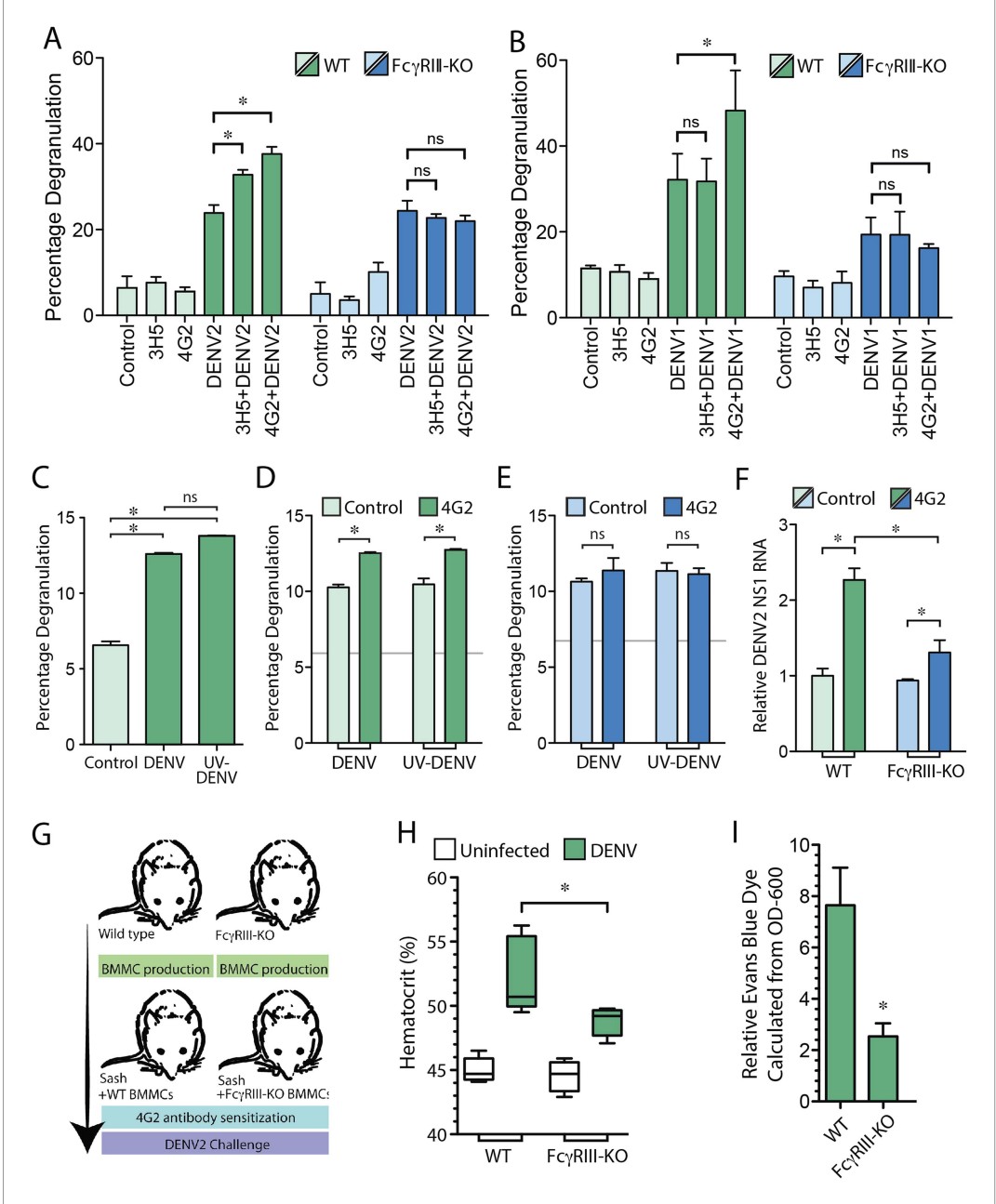

**Figure 4**. MC FcγRIII-induced degranulation promotes antibody-enhanced vascular leakage. WT and FcγRIII-KO BMMCs were pre-treated with antibodies 4G2 or 3H5, followed by exposure to (**A**) DENV2 or (**B**) DENV1 at an MOI of 1. Both 3H5 and 4G2 enhanced WT BMMC degranulation over DENV2 exposure alone in panel (**A**), but only 4G2 enhanced BMMC degranulation to DENV1 in panel (**B**). Both the 3H5- and 4G2-enhanced degranulation responses of BMMCs to DENV2, and the 4G2-enhanced degranulation responses to DENV1 were abrogated in FcγRIII-KO BMMCs. For (**A**–**B**), 3H5 or 4G2 pretreatment did not change degranulation levels of FcγRIII-KO BMMCs over DENV exposure alone. All statistical comparisons in (**A**–**B**) were performed by 1-way ANOVA with Bonferroni's post-test, *$p < 0.05$. (**C**) Degranulation is significantly induced by DENV and UV-DENV in mouse BMMCs. (**D**) Degranulation responses of WT BMMCs to both DENV and UV-DENV were increased significantly in the presence of antibody 4G2. (**E**) Antibody 4G2 did not enhance degranulation of FcγRIII-KO BMMCs in response to DENV or UV-DENV. In (**D**–**E**), baseline control levels are denoted by gray lines. (**F**) Both WT and FcγRIII-KO BMMCs showed increased levels of DENV2 replication in the presence of 4G2, compared to control BMMCs but the levels were reduced in 4G2-treated FcγRIII-KO BMMCs compared to WT BMMCs treated with 4G2. All comparisons designated by * were significantly different ($p < 0.05$), as determined by ANOVA with Tukey's multiple comparison test. (**G**) This panel depicts the

*Figure 4. continued on next page*

*Figure 4. Continued*

experimental design for BMMC adoptive transfer studies. MC-deficient Sash mice were reconstituted with BMMCs produced from either WT or FcγRIII-KO congenic controls. After maturation, all mice were pre-sensitized with 4G2 antibody prior to challenge with DENV2 ($1 \times 10^6$ pfu) by i.p. injection (or saline control, uninfected). At 24 hr after injection with saline or DENV2, vascular leakage was measured by (**H**) hematocrit and (**I**) EBD detection in the liver after saline perfusion, shown as relative detection, normalized to uninfected controls. (**G–I**) Mice reconstituted with FcγRIII-KO BMMCs showed reduced vascular leakage compared to mice reconstituted with WT BMMCs, demonstrating the role of MC-expressed FcγRIII in antibody-enhanced vascular leakage. Significance was determined in (**H**) by 1-way ANOVA with Bonferroni's post-test, and in (**I**) by un-paired Student's *t*-test; *$p < 0.05$.

We have previously shown that MC degranulation occurs independent of DENV replication (*St John et al., 2011*), and consistent results were obtained in this study, where there was not a significant difference between degranulation of BMMCs exposed to DENV or UV-inactivated DENV (*Figure 4C*). The DENV-specific antibody 4G2 also enhanced degranulation of WT BMMCs to a similar extent when exposed to either DENV or UV-inactivated DENV in the presence of 4G2 (*Figure 4D*). In contrast, FcγRIII-KO BMMCs, showed baseline degranulation responses, but no enhanced degranulation in the presence of specific antibody (*Figure 4E*). Although equivalent levels of degranulation occurred in response to live and UV-inactivated virus, we hypothesized that antibodies could potentially enhance replication of DENV within BMMCs through uptake by FcγRIII. The incubation time of antibodies and DENV with MCs was extended to 24 hr post-infection, and the relative amount of viral replication in BMMCs was determined by real time RT-PCR for the viral NS1 gene. Our results showed that both antibody-treated groups of WT, and FcγRIII-KO BMMCs had heightened levels of DENV NS1 compared to control cells, although 4G2-treated FcγRIII-KO BMMCs had reduced levels of DENV NS1 compared to 4G2-treated WT BMMCs (*Figure 4F*). These results demonstrate that FcγRIII can also contribute to ADE in BMMCs, but is likely not the only Fc receptor responsible for the enhanced replication that is observed in antibody-treated MCs. Consistent with previous findings, we found that UV-inactivated DENV efficiently induced degranulation, supporting that degranulation responses are uncoupled from viral replication. Furthermore, FcγRIII-deficiency reduces the magnitude of ADE in MCs but enhanced viral replication is not completely abrogated, while the antibody-dependent portion of the degranulation response is.

To address the role of FcγRIII on MCs during antibody-mediated enhancement of DENV-induced vascular leakage, in vivo, we reconstituted MC-deficient Sash mice with either WT BMMCs or BMMCs generated from mice deficient in FcγRIII, as summarized in *Figure 4G*. Mice were then injected with sensitizing antibody (4G2), followed by a DENV2 challenge 24 hr later. Again, we measured vascular permeability by hematocrit analysis and EBD injection followed by perfusion to quantitate vascular leakage in the liver, 24 hr post-infection. In support of a role for MC-expressed FcγRIII in vascular leakage, Sash mice repleted WT BMMCs displayed higher hematocrit levels (*Figure 4H*) and EBD leakage into the liver (*Figure 4I*) in response to DENV2 infection after 4G2 sensitization, compared to mice reconstituted with FcγRIII-KO BMMCs in response to DENV2 infection after 4G2 sensitization. These findings suggest that FcγRIII on MCs plays an essential role in the antibody-enhanced vascular leakage during DENV infection.

Monoclonal antibodies are frequently used to model the role of antibodies in secondary challenges since they are validated to be target-specific and can be delivered in precise quantities in vivo. However, since the serum antibody response to a primary infection is polyclonal, we also performed adoptive transfer experiments to validate the potential of pre-existing polyclonal serum antibodies to enhance vascular leakage in response to secondary DENV challenge in vivo. In brief, animals were infected with DENV2 and allowed to clear the primary infection, after which serum was harvested from the DENV2 post-immune animals and naïve controls 21 day post-infection. The post-immune or control serum was then adoptively transferred into naïve mice, which were subsequently challenged with an i.p. infection with either DENV1 or DENV2 (*Figure 5A*). The viral burden in the spleen, assessed by real time RT-PCR, 24 hr post-infection, was not found to be significantly different between mice that received control serum vs mice that received heterologous post-immune serum (*Figure 5B*). When the vascular leakage of infected mice was measured by hematocrit 24 hr post-infection, we observed that i.p. challenge with either DENV1 (a heterologous challenge) or DENV2

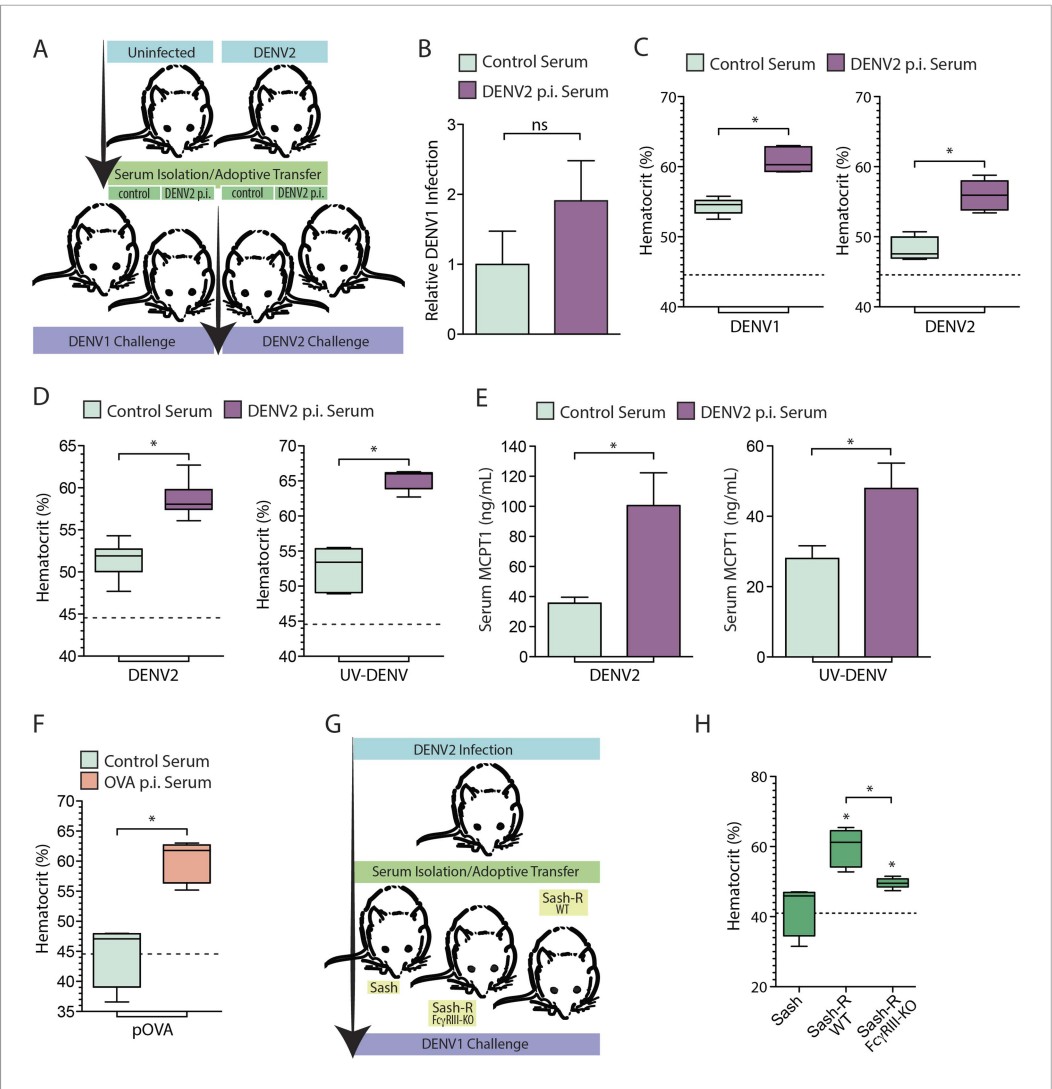

**Figure 5**. Vascular leakage in response to live and inactivated DENV after passive immunization. (**A**) A schematic depicting the experimental design of serum-adoptive transfer experiments where post-immune serum from DENV2-infected or naïve mice was adoptively transferred into naïve hosts that were then challenged with $1 \times 10^6$ pfu of either DENV1 or DENV2. (**B**) No significant (ns) differences were detected in the levels of DENV1 infection at 24 hr between groups infected after passive transfer of either control serum or serum from DENV2 post immune (p.i.) mice. Significance was determined by Student's un-paired $t$-test ($p = 0.2905$, $n = 5$). DENV1 RNA levels are displayed normalized to actin and the levels in the control group of mice. (**C**) Hematocrit analysis showed that vascular leakage was enhanced by either a heterologous challenge with DENV1 ($p = 0.0001$) or a homologous challenge with DENV2 ($p = 0.0005$), compared to mice provided control serum from naïve animals. For reference, the dashed line represents the baseline hematocrit value for naïve WT mice. (**D**) Similarly, mice challenged with UV-inactivated DENV (UV-DENV) showed elevated hematocrit values that were similar to mice given live virus. (**E**) Serum from control (naïve) mice or DENV2 p.i. mice was adoptively transferred to recipient mice. Levels of serum MCPT1 were measured by ELISA 24 hr after challenge of recipient mice with either DENV2 or UV-DENV2. Passive immunization with DENV2 p.i. serum resulted in elevated concentration of the MC activation biomarker, MCPT1, after challenge. For each panel, $p \leq 0.05$; $n = 5$. (**F**) Mice adoptively transferred serum from mice immunized against the antigen OVA or control serum were challenged with particulate OVA (pOVA). Antigen alone-treated mice did not have elevated hematocrit levels, but mice given OVA p.i. serum had significantly increased hematocrit levels at 24 hr. (**G**) The experimental design for adoptive transfer experiments where serum from DENV2 post immune animals was transferred into MC-deficient Sash mice, and Sash mice that had been reconstituted (Sash-R) with either FcγRIII-KO BMMCs or WT BMMCs, followed by a heterologous DENV1 challenge. (**H**) Mice reconstituted with WT MCs had significantly increased vascular leakage based on hematocrit analysis compared to Sash mice alone ($p = 0.002$), as did mice reconstituted with FcγRIII-KO BMMCs ($p = 0.047$); however, mice reconstituted with WT MCs showed

*Figure 5. continued on next page*

Figure 5. Continued

a further increase over Sash-R mice repleted with FcγRIII-KO BMMCs (p = 0.003). The dashed line represents the baseline average hematocrit value for naive Sash mice. Statistical comparisons in data-containing panels were performed using Student's unpaired *t*-test.

(a homologous challenge) enhanced vascular leakage in mice that had been adoptively transferred DENV2 post-immune serum, compared to mice adoptively transferred naïve serum (*Figure 5C*).

Our in vitro results suggested that MC degranulation in response to DENV is analogous to antigen-dependent MC degranulation, since degranulation was not dependent on viral replication (*Figure 4C–D*). Thus, we evaluated whether vascular leakage could be induced by UV-inactivated DENV in vivo and whether DENV post-immune serum also enhanced vascular leakage. Similar to mice injected with live DENV, inactivated DENV increased hematocrit levels at baseline, and the hemoconcentration was heightened in the presence of post-immune serum (*Figure 5D*). Concurrently, elevated levels of MCPT1, which is a biomarker for MC activation, could be detected in the serum of animals after both live and inactivated virus challenges (*Figure 5E*). Mice pre-treated with 4G2 had significantly higher serum levels of MCPT1 than mice given the live or UV-DENV challenges alone (*Figure 5E*). To compare this response to another antigenic challenge, we also adoptively transferred serum from mice immunized with ovalbumin (OVA) and subsequently injected the mice with particulate OVA (pOVA) as a challenge. The particulate form of the antigen was used since MC responses to particulate antigens are heightened compared to soluble antigens (*Jin et al., 2011*) and also because the size of DENV (∼50 nm) approximates the size of nanoparticles, rather than soluble proteins. As expected, mice that were challenged with the pOVA antigen alone did not have elevated hematocrit values at 24 hr, but mice that had been adoptively transferred OVA post-immune serum showed increased hematocrit values (*Figure 5F*), supporting that vascular leakage was increased in these animals.

Subsequently, we assessed the role of MC FcγRIII in contributing to the enhanced vascular leakage occurring after adoptive transfer of post-immune serum. For this, DENV2 post-immune serum was adoptively transferred into Sash mice that had been reconstituted with WT BMMCs and Sash mice that had been reconstituted with FcγRIII-KO BMMCs (*Figure 5G*). Similar to our findings after adoptive transfer of monoclonal antibodies (*Figure 4G–I*), FcγRIII on MC promoted increased vascular leakage during a secondary DENV challenge after post-immune serum transfer, since mice reconstituted with WT BMMCs had greatly enhanced vascular leakage responses over FcγRIII-KO BMMC-reconstituted mice (*Figure 5H*). Our results support that FcγRIII on MCs is critical for the enhanced vascular leakage observed after a heterologous DENV challenge.

## Discussion

We previously reported that DENV-induced MC degranulation during primary DENV infection results in the release of numerous MC products that can act on the vascular endothelium and influence the coagulation cascade (*St John, 2013*; *St John et al., 2013b*). Some of these products include the anti-coagulant heparin, MC-specific proteases, cytokines such as TNF, and other vasoactive factors. We previously detected the MC-specific product chymase at higher levels in the serum of DHF patients with secondary infection, compared to DHF patients experiencing primary infection (*St John et al., 2013b*). This suggested that MC activation (which is already elevated in primary DHF patients over control or DF patients [*St John et al., 2013b*]) is further heightened during secondary infection. Others also previously reported that DENV-post immune serum can prompt MC degranulation in response to DENV in vitro; however, it was assumed that this enhancement was due to the possibility that DENV-specific IgE was present in the serum (*Sanchez et al., 1986*), while any potential role of serum IgG in MC degranulation was not considered. In our current study, our data reveal that pre-existing IgG is able to enhance both MC degranulation and MC-dependent vascular leakage during DENV infection. We also found that MCs have equivalent degranulation responses to live and inactivated virus and that antibodies further enhance degranulation to live and inactivated virus similarly. Importantly, in vivo during a natural infection, both the concentration of antibodies and virus would vary along the course of infection and would presumably influence the degree of MC

degranulation and resulting vascular leakage. The observation that MC activation can be augmented by DENV-specific IgG constitutes a novel second mechanism of antibody-enhanced immune pathology during DENV infection that is dependent not on enhanced virus replication but on FcγR-mediated augmented MC degranulation responses.

Enhanced MC degranulation and MC-promoted vascular leakage in our system relied on the ability of pre-existing antibody to bind to DENV, since we saw that heterologous and homologous antibodies could enhance MC degranulation, but non-binding antibody (e.g., 3H5 which does not bind to DENV1) could not enhance degranulation. Increased vascular leakage in response to experimentally established systemic infection was not observed in Sash mice, which lack MCs, demonstrating that the enhanced vascular leakage was MC dependent. Furthermore, serum-adoptive transfer studies demonstrate that the mechanism of IgG-enhanced MC responses is dependent on MC-expressed FcγRIII. Importantly, FcγRIII-deficiency had no effect on direct MC degranulation responses to DENV, and only affected the antibody-enhanced component of the degranulation response. Reconstitution of Sash mice with WT, FcγRIII-expressing MCs promoted enhanced vascular leakage in vivo after either monoclonal antibody administration or adoptive transfer of post-immune serum, while vascular leakage in mice reconstituted with FcγRIII-KO MCs did not reach the same heightened levels. Interestingly, even homologous serum could enhance MC-dependent vascular leakage in our system, where a sufficient dose of virus is injected i.p. to establish a systemic infection. Others have also observed that homologous antibodies can also promote classical ADE responses involving enhanced replication (*Kliks et al., 1988*; *Yamanaka et al., 2008*). However, homologous secondary DENV infections have never been reported in humans and are assumed to be quickly cleared due to a strong and neutralizing adaptive immune response, without the development of symptoms (*St John et al., 2013a*). Therefore, it is likely that the virus would not achieve sufficient systemic infection levels to cause widespread antibody-dependent MC activation in the context of a naturally acquired secondary homologous infection. Taken together, these data establish the role of FcγRIII in immune detection of IgG-DENV immune complexes by MCs in vivo and the downstream vascular pathology that can result from MC activation.

Secondary infection with a heterologous strain of DENV is one of the major risk factors associated with the development of severe disease in human DENV patients. This is thought to be due to the potential of pre-existing antibodies to promote Fc-receptor-mediated endocytosis of DENV, allowing higher infection burden in cells, such as monocytes, that express Fc receptors (*Halstead, 2007*). Some data exist to support this mechanism since pre-existing antibodies have been shown to enhance infection in monocytes in the immunocompromised mouse model and human monocytes can also experience enhanced infection ex vivo when exposed to DENV in the presence of heterologous post-immune serum (*Halstead et al., 1973*; *Shresta et al., 2006*). Our data reveal a second mechanism through which pre-existing antibodies can contribute to vascular pathology, by increasing Fc-receptor-mediated MC degranulation.

Each animal model of DENV has caveats, and in this case, it is important to emphasize that FcγRIII is the only activating Fc receptor that has been identified on mouse MCs. Human MCs also express FcγRI as a second activating Fc receptor for IgG (*Malbec and Daëron, 2007*), and the potential exists that both could mediate antibody-enhanced MC degranulation responses to DENV in humans. Additionally, we have previously published that the WT mouse model sustains replicating viral infection with the clinical isolate used in this study, particularly in lymphoid organs such as the spleen, and is also detectable in the liver (*St John et al., 2013b*). However, limited viral replication may accentuate the direct effects of mature virus particles on immune cells, over the products that are generated during cytosolic replication of virus. Therefore, defining this new role for MCs in DENV infection also does not diminish the importance of viral replication to pathogenesis. The products produced by other infected and dying cell types are understood to be influential during DENV-induced immune pathology. Additionally, this WT mouse model displays increased vascular leakage after heterologous challenge; however, we have not observed the severe symptoms of frank hemorrhage or multi-organ failure that are characteristic of the approximately 1% of most severe DENV infections that are diagnosed as DHF/DSS.

IgG-mediated MC degranulation has not been studied in as great detail as IgE-dependent degranulation responses, but both contribute to vascular pathology in independent contexts, such as the Arthus reaction and anaphylaxis (*Vaz and Ovary, 1968*; *Hazenbos et al., 1996*; *Sylvestre and Ravetch, 1996*). Degranulation involves the release of the pre-stored products that are packaged

within the granular compartments, and the dominant products stored in granules are heparin and proteases (*Kunder et al., 2011*). Additional granule-associated mediators include β-hexosaminidase (the enzyme that is used to quantitate MC degranulation), histamine, TNF, and others (*Kunder et al., 2011*; *Kunder et al., 2009*). While both anaphylaxis and DENV infection can ultimately result in systemic dysregulation of the vascular system, shock, and multiple organ dysfunction, variation in the clinical presentation of these responses raises questions about the similarity between the two mechanisms. Certainly, the kinetics of viral infection is different from that of acute allergic responses involving vascular dysregulation. During infection, one would expect that ramping of antigen (in this case, virus) concentration in vivo would occur as infection spreads, in contrast to classical allergic responses, which are usually directed towards a bolus dose of exogenously derived antigen, such as food, drugs, or environmental substances. We expect that these differences in antigen dose, delivery, tissue distribution, and other factors could influence the kinetics and symptoms of vascular pathology and shock, in spite of both conditions involving a component of MC response in the underlying mechanism of pathology.

## Materials and methods

### Cell lines and virus strains

Rat basophilic leukemia-2H3 cells (RBLs) were cultured in α-MEM (Invitrogen, Life Technologies, Singapore) supplemented with 10%FBS, 100 U/ml penicillin, and 100 µg/ml streptomycin in a 5% $CO_2$ incubator at 37°C. Sub-culturing was performed using trypsin–EDTA (Invitrogen). For infections, DENV serotypes 1–4, clinical isolate strains Eden1, Eden2, Eden3, and Eden4 were used. These low-passage clinical isolates were obtained from the Duke-NUS reference laboratory and derived from the clinical study (Early Dengue Infection and Outcomes Study, Eden) (*Low et al., 2006*). Virus strains were propagated in *Aedes albopictus* C6/36 mosquito cells (CRL-1660; ATCC), maintained in RPMI medium 1640 with 25 mM HEPES, and titered using standard methods (*Schulze and Schlesinger, 1963*).

### Animal studies and infections

MC-deficient mice (Wsh/Wsh; 'Sash') and FcγRIII-deficient mice (*Fcgr3$^{tm1Sjv}$*) were originally purchased from Jackson Laboratories and bred in-house. Control mice on a C57BL/6 background were purchased from either Biological Resource Centre or InVivos (Singapore). DENV infection was initiated by i.p. injection of $1 \times 10^6$ pfu of DENV. To generate BMMCs for both in vitro experiments and in vivo reconstitution studies, bone marrow was flushed from mouse femurs and cultured in RPMI medium containing 10% FBS, 1% supernatant from Cho-KL cells which contains stem cell factor (produced in-house), penicillin and streptomycin, HEPES, trypsin inhibitor, sodium pyruvate (all from Invitrogen, Life Technologies, Singapore), and recombinant IL-3 (5 ng/ml, R&D Systems, Minneapolis, MN). After 4 weeks, MCs were verified to be >95% pure by toluidine blue (Sigma-Aldrich, Singapore) staining prior to use in other assays. Reconstitution of Sash mice was performed by injecting $1 \times 10^7$ BMMCs, i.v., followed by a 6-week period prior to use in experiments to allow full engraftment, as previously described (*Kunder et al., 2009*). For drug treatment studies, Cromolyn (Sigma-Aldrich) was given by a single i.p. injection at a dose of 3 mg/mouse. For sensitization of animals with monoclonal antibodies 4G2 or 3H5, 10 µg of antibody or isotype controls IgG1 or IgG2a (BD Pharmingen, Singapore) was injected in a 100 µl volume of PBS. Adoptive transfer experiments were performed by infecting mice with DENV2 or immunizing mice with 100 µg of OVA protein (both i.p.) and harvesting serum 3 weeks post-infection. Serum from post-immune or naïve control animals was pooled from multiple mice. Recipient mice were injected with 100 µl of serum and challenged 24 hr later with $1 \times 10^6$ pfu of either DENV1 or DENV2 by i.p. injection. All animal experiments were performed according to protocols approved by the SingHealth Institutional Animal Care and Use Committee.

### Measurement of MC degranulation

A standard β-hexosaminidase assay was used, as previously described (*St John, 2013*), to quantitate in vitro MC-degranulation (of either RBLs or BMMCs) in response to DENV at a MOI of 1. For some groups, prior to exposure to DENV, monoclonal antibodies 3H5 or 4G2 were incubated with MCs for 1 hr in cell culture, prior to addition of the DENV stimulus. To assess in vivo degranulation by flow cytometry, we performed intracellular staining experiments to quantify the levels of intracellular heparin, a MC-specific, granule-associated marker. Mice were infected by i.p. injection of $1 \times 10^6$ pfu

of DENV. After 24 hr, peritoneal lavage was obtained from individual mice in experimental and control groups and their peritoneal cells were prepared for flow cytometry. Cells were surface-stained for the MC marker ckit (CD117), followed by fixation and intracellular staining for MC granules using the granule heparin-specific probe avidin, according to a previously published method (*St. John et al., 2011*).

## Measurement of vascular leakage

Vascular leakage was assessed at the reported time points after infection using an EBD leakage assay, as previously described (*St John et al., 2013b*). In brief, EBD (Sigma-Aldrich) was prepared as a 1% solution in PBS, filtered and injected (100 µl volume) by tail vein, 30 min prior to euthanasia. Subsequently, the circulation of each animal was perfused fully with 15 ml of PBS. Biopsies of liver tissue were then obtained, and the mass of each biopsy was recorded prior to homogenization in PBS. After eliminating cellular debris by centrifugation the OD-600 of tissue supernatants was measured to quantify the amount of EBD leakage into tissues, based on a standard curve. EBD values were normalized to the tissue mass to give a concentration of EBD within tissues. To obtain hematocrit values, blood was collected in EDTA-tubes from individual mice via the maxillary vein at the designated time points after infection. Blood was run on an automatic hematology analyzer to obtain hematocrit values.

## Statistical analysis

Prism 5 and Excel were used to determine statistical significance. For comparisons of multiple groups, either 1-way or 2-way ANOVA was performed using Bonferroni's multiple comparison post-test to determine statistical significance amongst groups. Where indicated Student's un-paired $t$-test was used to evaluate differences between two groups. Data were considered significant at $p \leq 0.05$.

## Acknowledgements

This work was funded by NMRC/NIG/1053/2011 and start-up funds from Duke-NUS Graduate Medical School. The authors thank Abhay PS Rathore for critical manuscript review.

## Additional information

### Funding

| Funder | Grant reference | Author |
| --- | --- | --- |
| National Medical Research Council (NMRC) | 1054/2011 | Ashley L St John |
| Duke-NUS Graduate Medical School | Start-up Funding | Ashley L St John |

The funders had no role in study design, data collection and interpretation, or the decision to submit the work for publication.

### Author contributions

AS, CJJ, SABA, Manuscript review, Acquisition of data, Analysis and interpretation of data; AS, Acquisition of data, Analysis and interpretation of data, Drafting or revising the article; ALSJ, Conception and design, Acquisition of data, Analysis and interpretation of data, Drafting or revising the article

### Ethics

Animal experimentation: All animal experiments were performed according to protocols approved by the Duke-NUS/SingHealth IACUC (Protocols #SHS/710 and #SHS/774).

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
