## [Decision Letter]

Thank you for sending your work entitled “Dengue vascular leakage is augmented by Fcγ receptor-mediated mast cell degranulation” for consideration at *eLife*. Your article has been favorably evaluated by Prabhat Jha (Senior editor), a Reviewing editor, and two reviewers.

The Reviewing editor and the reviewers discussed their comments before we reached this decision, and the Reviewing editor has assembled the following comments to help you prepare a revised submission.

This manuscript shows that DENV specific antibodies primed on mast cells (MCs), after binding to DENV can trigger MCs to degranulate and lead to the development of vascular leakage in mice. These findings potentially offer an alternative role of antibody-dependent-enhancement (ADE) induced plasma leakage in dengue hemorrhagic fever (DHF). The manuscript is very straightforward and the results are quite clear. However, there are points that need clarification in terms of their relevance to DHF in humans.

Major comments:

1) MC degranuation can be enhanced in the presence of pre-existing homologous or heterologous DENV antibodies. In humans, DHF usually occurs after a secondary infection with a different virus serotype. How would the authors explain the relevance of homotypic antibodies enhancing vascular leakage in mice (Figure 3)? This is a very important point that needs to be addressed. A second infection by the same virus serotype does not lead to DHF in humans.

2) Similar to point 1: In serum transfer experiment (Figure 4), how would the authors explain that both homotypic and heterotypic secondary infections show similar results in vascular leakage (unlike humans)?

3) It seems that MC degranulation can occur independently of active DENV replication. In addition, DENV antibody tends to increase RNA replication in MCs as shown by higher NS1 transcript. However, discussion on the mechanism of the linkage between DENV infection (replication) and MC degranulation in the presence of specific antibody is missing in the manuscript. If MC degranulation is not dependent on infection, how would the authors explain the phenomenon of enhanced degranulation and vascular leakage in mice induced by DENV specific Abs? Can antibodies specific to UV-irradiated DENV (no capability of replication) also enhance MC degranulation and vascular leakage in vivo?

4) Related to point 4 (as pointed out by the authors), the interaction between DENV and its specific Abs enhance viral replication in MCs, in all in vivo experiments, the authors should also show the data of viral burden in the blood and major target organs like liver and spleen which are important as they may help to explain the enhanced pathology induced by DENV specific Abs in this mouse model. This is despite the fact that MC degranulation does not depend on active viral replication in MCs, as the two events might not have a direct causal connection.

5) Would any antigens bound to specific Abs on MCs trigger vascular leakage in mice? If yes, how do the key findings described in the manuscript specifically explain ADE (of vascular leakage) in DHF? It would be useful to also do experiments in which mice are inoculated by antigens other than DENV following the injection of its specific Abs and look to see if the signs of vascular leakage similar to that was observed in DHF also occur. It is important to know whether or not replication of DENV is necessary for the development of vascular leakage in this ADE model.

6) We assume that mice don't get severe DHF after secondary heterotypic infection like humans so that serial infection with a different serotype of DENV might not give DHF picture. This point should be discussed.

7) Figures 1 and 2: the authors used peritoneal MCs, would this represent an MC population that could cause an increase in vascular leakage?

8) A previous study by the group demonstrated that DENV recognition by mast cells causes chimase, lipid mediators and TNF release. Interestingly, cysteinil leukotrienes (accessed using the antagonist Montelukast) but not TNF were important to vascular leakage. What mediators are released upon antibody-dependent FcγRIII activation and what mediators contribute to the increased vascular leakage? I think that this issue would contribute to the understanding of FcγRIII activation on mast cell in general and on dengue pathogenesis in particular, thus increasing the importance and novelty of the study.

Minor comments:

The authors claimed that “Others also previously reported that DENV-post immune serum can prompt MC degranulation in response to DENV in vitro; however, it was assumed that this enhancement was due to the possibility that DENV-specific IgE was present in the serum (31), while any potential role of serum IgG was not considered.” Albeit mast cell degranulation was not directly tested, a previous article demonstrated the role of FcγRIII in antibody-enhanced dengue virus infection of human mast cells and increased chemokine release (Brown et al., J Leuk Biol, 2006). This work should be cited.

---

## [Author Response]

*This manuscript shows that DENV specific antibodies primed on mast cells (MCs), after binding to DENV can trigger MCs to degranulate and lead to the development of vascular leakage in mice. These findings potentially offer an alternative role of antibody-dependent-enhancement (ADE) induced plasma leakage in dengue hemorrhagic fever (DHF). The manuscript is very straightforward and the results are quite clear. However, there are points that need clarification in terms of their relevance to DHF in humans*.

We thank the reviewers for their thorough reading of our manuscript. We have addressed their questions by providing a few new figure panels and improving the Discussion and Introduction sections. Since panels were added to Figure 4, it has been split into two figures to improve readability.

*Major comments*:

*1) MC degranuation can be enhanced in the presence of pre-existing homologous or heterologous DENV antibodies. In humans, DHF usually occurs after a secondary infection with a different virus serotype. How would the authors explain the relevance of homotypic antibodies enhancing vascular leakage in mice (*Figure 3*)? This is a very important point that needs to be addressed. A second infection by the same virus serotype does not lead to DHF in humans*.

Homologous secondary DENV infections have never been reported in humans. We assume that they are quickly cleared without the development of symptoms as a result of an effective neutralizing adaptive immune response (involving both antibody and memory T cell responses). Therefore, in agreement with the reviewer, we don’t expect that virus would achieve sufficient systemic infection levels to cause widespread antibody-dependent MC activation in the context of a naturally acquired secondary homologous infection. In this animal model, we have given a homologous challenge in mice that were adoptively transferred homotypic antibodies. The challenge is also given by i.p. injection in order to achieve a systemic infection.

This model allows us to tease apart the contributions of pre-existing antibody and viral particles to vascular leakage by demonstrating that antibody cross-linking on mast cells is sufficient to induce vascular leakage. In this situation, the antigen-antibody complexes activate mast cells via FcγRIII. Similarly, others have shown that ADE of monocytes can occur in the presence of homologous antibody ([44]; Kliks, 1988). These two types of experiments examining the influence of homologous antibody on either ADE (idem) or MC degranulation (presented here) show that antibody binding is critical for the interactions of immune cells with virus, even though other factors (such as effective immune clearance) contribute to whether substantial amounts of immune complexes are able to interact with immune cells in vivo. The Discussion has been updated to include this comment.

*2) Similar to point 1: In serum transfer experiment (*Figure 4*), how would the authors explain that both homotypic and heterotypic secondary infections show similar results in vascular leakage (unlike humans)*?

Similar to the response for question #1, we believe that multiple factors contribute to quick clearance of homotypic but not heterotypic dengue infections in humans. In this model, when mice experience a secondary infection with DENV2 in the presence of antibody, vascular leakage increases. (The figure in question is now renamed Figure 5.)

*3) It seems that MC degranulation can occur independently of active DENV replication. In addition, DENV antibody tends to increase RNA replication in MCs as shown by higher NS1 transcript. However, discussion on the mechanism of the linkage between DENV infection (replication) and MC degranulation in the presence of specific antibody is missing in the manuscript. If MC degranulation is not dependent on infection, how would the authors explain the phenomenon of enhanced degranulation and vascular leakage in mice induced by DENV specific Abs? Can antibodies specific to UV-irradiated DENV (no capability of replication) also enhance MC degranulation and vascular leakage in vivo*?

The further discussion that the reviewer requested has been added to the Discussion section. We should clarify that the data in Figure 5 showed a trend towards increased NS1 transcript in the spleen in animals given a heterologous challenge, but this was less than 2-fold and not significant in our data set.

The reviewer makes an interesting point that antibodies should also enhance mast cell degranulation in response to UV-inactivated virus. We have now performed several experiments to address this question. First, DENV and UV-DENV both cause significant and similar magnitude of degranulation of MCs in cell culture (Figure 4). Degranulation of MCs in response to both DENV and UV-DENV is enhanced by specific antibody (Figure 4), and this enhancement is dependent on FcγRIII (Figure 4).

We also examined the potential of UV-inactivated virus to induce vascular leakage in vivo. Our findings showed that, indeed, mice given UV-DENV develop hemoconcentration at 24h, which is enhanced by DENV-specific post-immune serum (Figure 5). Since DENV replicates in vivo but UV-DENV does not, the number of virus particles is expected to be higher in mice administered DENV than UV-DENV, subsequent to the first round of replication. Thus, later time points beyond the early hours of infection would not be appropriate for comparison.

*4) Related to point 4 (as pointed out by the authors), the interaction between DENV and its specific Abs enhance viral replication in MCs, in all in vivo experiments, the authors should also show the data of viral burden in the blood and major target organs like liver and spleen which are important as they may help to explain the enhanced pathology induced by DENV specific Abs in this mouse model. This is despite the fact that MC degranulation does not depend on active viral replication in MCs, as the two events might not have a direct causal connection*.

We previously reported that the WT mouse model sustained highest levels of replication in the spleen, thus, this organ was chosen to measure DENV replication for this follow-up study to advance our original findings. Due to the importance of this figure, we have now moved it from supplemental to Figure 5. We also have provided new data regarding the viral replication in the presence of the monoclonal antibody 4G2 (Figure 1). Furthermore, we examined virus replication in WT and Sash mice, Sash-R WT and Sash-R FcγRIII-KO, after adoptive transfer of either control or DENV-post immune serum. This experiment captures all of the key experimental variables that are relevant to the study, but again, we did not observe any significant differences in the viral burden amongst groups (Figure 6).Author response image 1.DENV burden in the spleen of MC-deficient and -repleted mice. Spleens were isolated 24h after infection with DENV and real time RTPCR was performed to quantify the viral burden in this DENV-target organ. In contrast to the significant differences in vascular leakage, viral burden did not differ significantly for the same number of mice (n=5; p=0.13).

*5) Would any antigens bound to specific Abs on MCs trigger vascular leakage in mice? If yes, how do the key findings described in the manuscript specifically explain ADE (of vascular leakage) in DHF? It would be useful to also do experiments in which mice are inoculated by antigens other than DENV following the injection of its specific Abs and look to see if the signs of vascular leakage similar to that was observed in DHF also occur. It is important to know whether or not replication of DENV is necessary for the development of vascular leakage in this ADE model*.

We have performed this experiment and found that antibody and antigen alone is sufficient to induce vascular leakage in mice (Figure 5). These findings are consistent with other literature examining mast cells, where antigen and antibody are sufficient to induce MC activation and related pathology in similar models (Vaz et al., 1969; Vaz et al., 1968; [4]). When mice are passively transferred post-immune serum of OVA-immunized mice, followed by challenge with particulate OVA, this also induces vascular leakage in mice (although unlike DENV and UV-DENV, pOVA does not elevate the vascular leakage at baseline; Figure 5). Interestingly, DENV is approximately 50nm in size so it would be more analogous to a particulate rather than soluble antigen. MCs have been shown to have enhanced degranulation responses to particulate antigens compared to soluble antigens (17).

*6) We assume that mice don't get severe DHF after secondary heterotypic infection like humans so that serial infection with a different serotype of DENV might not give DHF picture. This point should be discussed*.

It is correct that the symptoms of vascular leakage in the mouse model are enhanced by antibody but those symptoms do not meet the diagnostic criteria for humans with “DHF” or “severe dengue”. This point is discussed in the Discussion section.

*7)*
Figures 1 and 2*: the authors used peritoneal MCs, would this represent an MC population that could cause an increase in vascular leakage*?

MCs are distributed throughout connective tissues and many are located in the peritoneal cavity where the initial inoculating dose of DENV is given. Our experiments show that many MCs in the peritoneal cavity are activated, but we cannot exclude the possibility that MCs in other locations are activated as well, particularly since replicating viral infection is detected in the spleen and liver by 24h in this model (Figure 2 of St. John et al., eLife, 2013).

*8) A previous study by the group demonstrated that DENV recognition by mast cells causes chimase, lipid mediators and TNF release. Interestingly, cysteinil leukotrienes (accessed using the antagonist Montelukast) but not TNF were important to vascular leakage. What mediators are released upon antibody-dependent FcγRIII activation and what mediators contribute to the increased vascular leakage? I think that this issue would contribute to the understanding of FcγRIII activation on mast cell in general and on dengue pathogenesis in particular, thus increasing the importance and novelty of the study*.

Mast cell granule-associated mediators are pre-synthesized and stored in their granules, not produced de novo. In this paper, we have shown that the magnitude of degranulation responses is enhanced by FcγRIII activation (Figure 4), but we don’t expect that the calcium flux-dependent processes initiated by FcγRIII would result in different mediators being released. FcγRIII may activate other signaling events beyond degranulation. The extent to which this occurs and whether it influences the production of de novo synthesized mediators could be examined in future studies.

*Minor comments*:

*The authors claimed that* “*Others also previously reported that DENV-post immune serum can prompt MC degranulation in response to DENV in vitro; however, it was assumed that this enhancement was due to the possibility that DENV-specific IgE was present in the serum (*[31]*), while any potential role of serum IgG was not considered.*” *Albeit mast cell degranulation was not directly tested, a previous article demonstrated the role of* FcγRII *in antibody-enhanced dengue virus infection of human mast cells and increased chemokine release (Brown et al., J Leuk Biol, 2006). This work should be cited.*

It is important to note that the cytokine CCL5 is de novo transcribed in response to infection, and has not ever been shown to be granule-associated in MCs. Additionally, the KU812 cell line used in this report is described by the authors as a MC line but, rather, it is considered to be an immature pre-basophilic cell line (Yamashita et al, 2001; Sechet et al, 2003; Masuko et al, 1999; Kishi et al, 1984). Since an article by the same group first reported cytokine production by DENV infected cells that have phenotypic characteristics similar to mast cells (18), and that article more correctly represents the cell type in question, we have included this citation to establish the point raised by the reviewer.

The line mentioned above has also been updated to read “any potential role of serum IgG in MC degranulation was not considered”, to clarify that the Introduction section is specifically meant to discuss the mechanism of MC degranulation during dengue infection within the study by Sanchez et al.

*References*:

Yamashita M, Ichikawa A, Katakura Y, Mochizuki Y, Teruya K, Kim EH, et al. Induction of basophilic and eosinophilic differentiation in the human leukemic cell line KU812. Cytotechnology 2001, 36(1-3): 179-186.

Sechet B, Meseri-Delwail A, Arock M, Wijdenes J, Lecron JC, Sarrouilhe D.Immunoglobulin D enhances interleukin-6 release from the KU812 human prebasophil cell line. General physiology and biophysics 2003, 22(2): 255-263.

Masuko M, Koike T, Toba K, Kishi K, Kuroha T, Furukawa T, et al. Expression of eosinophil peroxidase in the immature basophil cell line KU812-F. Leukemia research 1999, 23(2): 99-104.

Kishi K, Takahashi M, Aoki S, Nagai K, Hirosawa H, Koike T, et al. A new Ph1 positive cell line (KU812) from a patient with blastic crisis of chronic myelogenous leukemia. Nihon Ketsueki Gakkai zasshi: journal of Japan Haematological Society 1984, 47(3): 709-718.